

# Stroke-related length of hospitalization trends and in-hospital mortality in Peru

L. Max Labán-Seminario[1], Rodrigo M. Carrillo-Larco[1,2] and Antonio Bernabé-Ortiz[1,3]

[1] CRONICAS Center of Excellence in Chronic Diseases, Universidad Peruana Cayetano Heredia, Lima, Peru
[2] Department of Epidemiology and Biostatistics, School of Public Health, Imperial College London, London, United Kingdom
[3] Universidad Científica del Sur, Lima, Peru

Corresponding author
Antonio Bernabé-Ortiz,
antonio.bernabe@upch.pe

## ABSTRACT

**Background:** Peru faces challenges to provide adequate care to stroke patients. Length of hospitalization and in-hospital mortality are two well-known indicators of stroke care. We aimed to describe the length of stay (LOS) of stroke in Peru, and to assess in-hospital mortality risk due to stroke, and subtypes.

**Methods:** This retrospective cohort study used hospitalization registries coding with ICD-10 from 2002 to 2017 ($N = 98,605$) provided by the Ministry of Health; in-hospital mortality was available for 2016–2017 ($N = 6,566$). Stroke cases aged ≥35 years were divided into subarachnoid hemorrhage (I60), intracerebral hemorrhage (I61), cerebral infarction (I63), and stroke not specified as hemorrhage or infarction (I64). Data included stroke LOS and in-hospital mortality; socio-demographic and clinical variables. We fitted a region- and hospital level-stratified Weibull proportional hazard model to assess the in-hospital mortality.

**Results:** The median LOS was 7 days (IQR: 4–13). Hemorrhagic strokes had median LOS longer than ischemic strokes and stroke not specified as hemorrhage or infarction ($P = <0.001$). The case fatality rate (CFR) of patients with stroke was 11.5% (95% CI [10–12%]). Subarachnoid hemorrhage (HR = 2.45; 95% CI [1.91–3.14]), intracerebral hemorrhage (HR = 1.95; 95% CI [1.55–2.46]), and stroke not specified as hemorrhage or infarction (HR = 1.45; 95% CI [1.16–1.81]) were associated with higher in-hospital mortality risk in comparison to ischemic strokes.

**Discussion:** Between 2002 and 2017, LOS due to stroke has not changed in Peru in stroke patients discharged alive. Hemorrhagic cases had the longest LOS and highest in-hospital mortality risk during 2016 and 2017. The findings of our study seem to be consistent with a previous study carried out in Peru and similar to that of HIC and LMIC, also there is an increased median LOS in stroke cases managed in specialized centers. Likewise, LOS seems to depend on the type of stroke, where ischemic stroke cases have the lowest LOS. Peru needs to improve access to stroke care.

## INTRODUCTION

Over the last decades, stroke has mounted to be among the top causes of death globally with 6.2 million fatal cases in 2017 (*GBD 2017 Causes of Death Collaborators, 2018*); of

note, 71% of them occurred in low- and middle-income countries (LMIC) (*Feigin et al., 2014*). In Latin America, the hemorrhagic cases seem to happen more often than in High Income Countries (HIC) (*Feigin et al., 2009*), which can lead to a large hospital burden of 7,632 stroke hospitalizations reaching 47,200 United States Dollars (USD) as 1-year hospital cumulative cost (*Ohinmaa et al., 2016*; *Kaur et al., 2014*; *Wang et al., 2014*; *Reed et al., 2001*) between 2008 and 2009. Likewise, the care cost, LOS, and mortality of hemorrhagic stroke are greater than ischemic stroke, which contribute to significant socioeconomic disparities in stroke care of universal healthcare systems, especially in scarce settings and in low socioeconomic status population (*Reed et al., 2001*; *Bray et al., 2018*; *Cesaroni et al., 2009*).

Although an emerging economy, Peru faces difficulties to provide adequate care to stroke patients because of the dearth of logistics and human resources as well as inequalities (*Ferri et al., 2011*) in the clinical settings of the Ministry of Health (MoH), which is organized into three levels of healthcare according to whether general or specialized care is required, such as the third-level facilities offer better support for stroke specialized care (*Ministerio de Salud, 2011*; *Sequeiros-Chirinos et al., 2020*). However, the most care facilities do not have sufficient capacity for emergency treatment, and stroke care centers with neuro-imaging are centralized and reduced (*Davalos & Málaga, 2014*; *Ministerio de Salud, 2020*). Stroke profiles have been studied in limited populations and scenarios in Peru (*Abanto et al., 2013*; *Jaillard, Hommel & Mazetti, 1995*; *Alvarado-Dulanto et al., 2015*; *Castañeda-Guarderas et al., 2011*). Likewise, the stroke mortality in Peruvian public hospitals decreased between 2005 and 2015 (*Atamari-Anahui et al., 2019*). However, an approximation of the use of hospital resources at the national level is not yet available. LOS provides general information about the efficiency of the use of hospital resources, and it is frequently reported in stroke quality registries (*Langhorne et al., 2020*). This study aims to describe LOS in Peru between 2002 and 2017, overall and by type of stroke; also, to study in-hospital mortality between 2016 and 2017.

## MATERIALS AND METHODS

### Study design

We conducted a retrospective cohort using national open access registries of the Peruvian Ministry of Health (MoH) aimed to determine if hemorrhagic stroke is associated with a longer hospital stay and the risk of in-hospital mortality compared to ischemic stroke and stroke not specified. MoH provided registries of cerebrovascular diseases with ICD-10 codes. Methods of collected data were detailed in *Labán (2021)*. Further descriptions are included in the Supplemental Methods.

### Study population

Participants were included using the methods described above in *Labán (2021)*. Stroke patients cared by physicians with an only primary diagnostic code ICD-10 were identified from first-, second- and third-level health facilities, which cover approximately 55% of Peru's population (*Seguro Integral de Salud, 2017*). The ICD-10 codes were standardized in all health facilities of MoH from 2002 onwards (*Ministerio de Salud, 2002*). We cleaned the

data following these steps. First, we selected the stroke diagnosis of interest: subarachnoid hemorrhage (ICD-10 code I60), intracerebral hemorrhage (ICD-10 code I61), cerebral infarction (ICD-10 code I63), and stroke not specified as hemorrhage or infarction (ICD-10 code I64). Second, we only kept patients who were discharged alive by a physician from 2002 to 2017 to analyze stroke LOS (*i.e.*, patients whose discharge record was due to other reasons for discharge, such as voluntary discharge, transferred patients, discharge with death registry, and missing data were not included due to possible biases related to the course of in-hospital medical management); on the other hand, stroke deaths were only available in 2016 and 2017 which were included for in-hospital mortality analysis. Third, we only included adults aged ≥35 years because only a relatively small proportion of subjects (5.6%) were younger than 35 years in the complete dataset; and more than 60% of them did not have discharge date data to calculate the length of stay and the most frequent cause of the cerebrovascular disease belongs to the group of other cerebrovascular diseases (ICD-10 code I67) with 24% ($N = 1,730$). Lastly, we excluded cases with LOS below and above the 1st and 99th percentile of each year to avoid implausible values or potential data entry errors (*Wang et al., 2014*).

## Variables

The outcomes of interest were: (i) stroke LOS, defined as the days between discharge and hospitalization date, for which we employed data from 2002 to 2017. (ii) In-hospital mortality (time-to-event variable), which was defined as the days elapsed between hospital admission date until the stroke case was discharged by medical indication (censored) or died during hospitalization, where we included all data related to time to death or discharge from 2016 to 2017.

We used socio-demographic and clinical variables to describe LOS trends and in-hospital mortality: sex; age (35–54, 55–74 and ≥75 years); Peruvian region (Lima/Callao, rest of the Coast, Highlands and Amazon) and health facility level (level I: primary care facilities; level II: intermediate care facilities; and level III: specialized hospitals and institutes), employed to fit the mortality analysis; and type of stroke, which was the exposure of interest (ICD-10: subarachnoid hemorrhage (I60), intracerebral hemorrhage (I61), cerebral infarction (I63) used as a reference, and stroke not specified as hemorrhage or infarction (I64)).

## Statistical analysis

Statistical methods used in this study have been detailed above in *Labán (2021)*. We described variables through mean/median or absolute/relative frequencies, according to the distribution and whether they were numerical or categorical. We computed medians along with interquartile ranges (IQR) because LOS had a non-normal distribution using the Kolmogorov-Smirnov test. We carried out comparisons between variables using the following tests: the Chi-square test compared categorical variables, whereas the Kruskal-Wallis test and corrected for multiple comparisons (Dunn's test with Bonferroni correction) for continuous numerical variables. Also, we employed the Mann-Whitney-Wilcoxon Rank Sum test to compare the means of LOS according to sex and level.

We analyzed the trend of LOS from 2002 to 2017 by sex through the Spearman correlation (rho). We used the Poisson distribution with 95% confidence intervals (95% CI) to estimate the case-fatality rate (CFR), and we employed the two-proportion z-test to get p-values. For the mortality analysis, we used the Kaplan-Meier method (unadjusted) and log-rank test to examine the in-hospital survival by type of stroke. We evaluated the proportional hazard assumptions for the primary exposure as well as for the additional covariates (age and sex), then we fitted a region- and hospital level-stratified Weibull proportional hazard model. We employed the Schoenfeld test to corroborate the proportional hazard assumption, and we evaluated the Weibull distribution of the baseline hazard through the log(-log(Survival)) and log(time) plot. We analyzed interactions in the model described above through the likelihood-ratio test: stroke type and level of the facility, as well as stroke type and regions. We developed the analyses with R (v4.0.0) (*R Core Team, 2019*).

### Ethics

This study is an analysis of health records with de-identified information. The analyzed data did not present any personal information, and no human subjects were enrolled. We did not seek ethical approval by an institutional review board or ethics committee.

## RESULTS

### Overall description

Between 2002 and 2017, a total of 98,605 MoH records from first-, second- and third-level healthcare facilities were identified. Of these, 58% ($N$ = 57,153) were cerebrovascular diseases of interest (hemorrhagic stroke, ischemic stroke, and stroke not specified). The most frequently excluded diagnoses were: other specified cerebrovascular diseases (ICD-10 code I678) with 20% ($N$ = 19,962) and cerebrovascular disease unspecified (ICD-10 code I679) with 12% ($N$ = 11,912). Only 39,810 records were included in the final analysis (Fig. 1). About 45% ($N$ = 17,815) of stroke cases were from first- and second-level facilities; 50% ($N$ = 19,840) of cases were reported from Lima/Callao, followed by the rest of the Coast (*i.e.*, outside Lima, the capital city) with 27% ($N$ = 10,851).

Cases of stroke aged, on average, 68.1 (SD = 13.9) years. Between 2002 and 2014, there were apparently more women than men (14,312 *vs* 13,741); after 2014, the number of cases of men increased slightly that of women (6,055 *vs* 5,702). Between 2002 and 2017, most of the stroke cases were not specified as hemorrhage or infarction (51%, $N$ = 20,449), followed by cerebral infarction (24%, $N$ = 9,635), intracerebral hemorrhage (16%, $N$ = 6,352), and subarachnoid hemorrhage (8%, $N$ = 3,374). Both first- and second-level facilities reported 64% ($N$ = 13,117) of stroke cases not specified as hemorrhage or infarction, whereas the third-level facilities reported 76% ($N$ = 7,382) of hemorrhagic cases.

### Length of hospitalization

Stroke cases had a median LOS of 7 days (IQR: 4–13) ranging from 2 to 55 days, and it did not vary across years (Fig. 2A). Between 2002 and 2017, there was no clear LOS difference by sex ($P$ = 0.05): men (7 (IQR: 4–13)) and women (7 (IQR: 4–13)) (Fig. 2B), and there

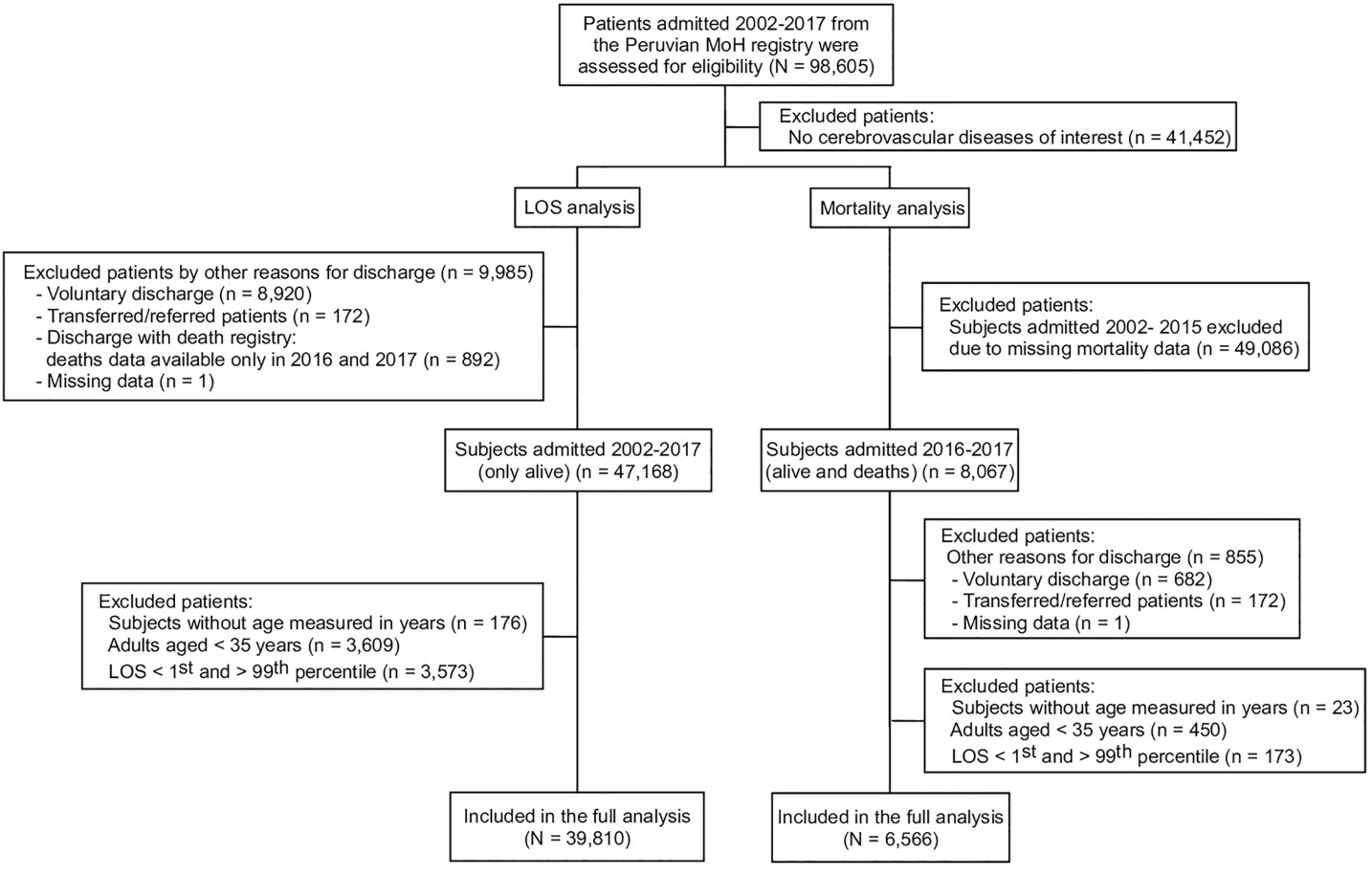

**Figure 1** Flowchart of patients from the Peruvian MOH admitted between 2002 and 2017 and assessed for eligibility.

were differences of LOS in each group of age ($P < 0.001$): <55 (8 (IQR: 5–15)), 55–74 (7 (IQR: 4–13), and 75+ (7 (IQR: 4–12))); that is, older people would stay fewer days than younger patients (Fig. 2C).

Subarachnoid hemorrhage showed the largest LOS (Fig. 2D). Hemorrhagic conditions had longer LOS than the ischemic cases and stroke not specified as hemorrhage or infarction ($P < 0.001$): subarachnoid hemorrhage (12 (IQR: 6.25–20)), intracerebral hemorrhage (10 (IQR: 6–17)), cerebral infarction (8 (IQR: 5–14)) and stroke not specified as hemorrhage or infarction (6 (IQR: 4–10)). Across years, third-level facilities had higher LOS than first- and second-level facilities ($P < 0.001$): I–II (5 (IQR: 3–9)) and III (10 (IQR: 6–16)) (Fig. 2E); and Lima/Callao and the rest of the Coast were the regions with the largest LOS ($P < 0.001$): Lima/Callao (9 (IQR: 5–16)), Rest Coast (7 (IQR: 4–11)), Amazon (5 (IQR: 3–7)) and Highlands (6 (IQR: 3–9)) (Fig. 2F).

## Mortality and type of stroke

Between 2016 and 2017, 6,566 cases of stroke were recorded, and 757 deaths occurred during that period (CFR = 11.5%; 95% CI [10.7–12.4%]). Females had larger CFR than males: 13% (95% CI [12–14%]) *vs* 10% (95% CI [9–11%], $P < 0.001$). Subarachnoid

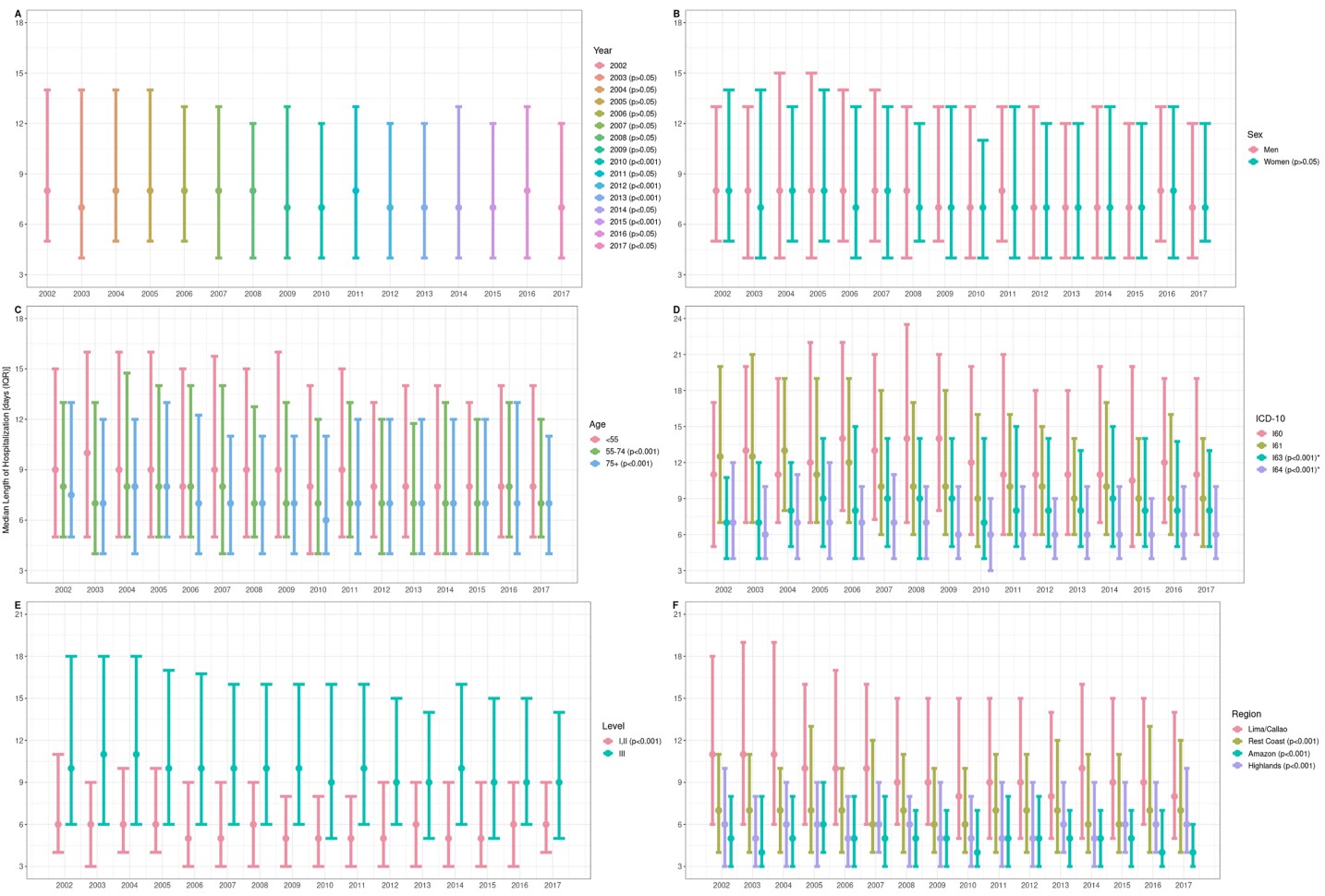

**Figure 2 Length of hospitalization due to cerebrovascular disease in Peru 2002–2017, by (A) year, (B) sex, (C) age group, (D) stroke type, (E) care level, and (F) region.** LOS data were represented through median and IQR. *P*-values for LOS comparisons by sex and care level were performed using Wilcoxon-rank sum. *P*-values for multiple LOS comparisons by year, age, ICD-10 and region were obtained using the Wilcoxon rank-sum and Dunn's test following Kruskal-Wallis test. *P*-value obtained from the comparison between hemorrhagic condition group (I60 and I61), ischemic cases (I63), and stroke not specified (I64) used Kruskal-Wallis test. Y-axis: (A) years of hospitalization of stroke patients discharged alive. (B) Sex of the stroke cases discharged alive. (C) Age in years of the stroke cases discharged alive. (D) International Classification of Diseases (ICD-10) of the stroke cases discharged alive. (E) Category of the health facility corresponding to the stroke cases discharged alive. (F) Region of the health facilities where the stroke cases discharged alive were managed. 

hemorrhage (CFR = 21.5%; 95% CI [18–25%]) and intracerebral hemorrhage (CFR = 14.9%; 95% CI [12.8–17%]) had larger CFR than stroke not specified as hemorrhage or infarction (CFR = 10.5%; 95% CI [9.3–11.7%]) and cerebral infarction (CFR = 6.9%; 95% CI [5.7–8.1%]). Clinical and socio-demographic variables according to mortality status are described in Table 1.

Kaplan-Meier estimates showed an in-hospital survival difference among the stroke subtypes (Fig. 3; log-rank test *P* < 0.001). Patients with ischemic stroke had the highest survival compared with other types of stroke. Violations of the proportional hazard (PH) assumption in the crude (*P* = 0.004) and adjusted (*P* = 0.006) models were found, and baseline hazards seemed to have a linear relationship between log(−log(survival)) and log (time). Likewise, the global Schoenfeld Test was statistically significant (*P* = 0.004). As a

**Table 1 Socio-demographic and clinical variables according to the outcome of hospitalization between 2002 and 2017.**

| Socio-demographic and clinical characteristics | Admissions to hospitalization between 2002 and 2017 | | | P value[c] |
|---|---|---|---|---|
| | Patients admitted 2002–2017 for LOS analysis[a] | Patients admitted 2016–2017 for intra-hospital mortality analysis[b] | | |
| | Discharged alive (N = 39,810) | Discharged alive (N = 5,809) | Deaths (N = 757) | |
| Sex | | | | |
| Men | 19,796 (49.7%) | 2,974 (51.2%) | 331 (43.7%) | <0.001 |
| Women | 20,014 (50.3%) | 2,835 (48.8%) | 426 (56.3%) | |
| Age (years) (mean ± SD) | 68.1 ± 13.9 | 68.3 ± 13.8 | 70 ± 14.2 | |
| Age (years) | | | | |
| <55 | 7,394 (18.6%) | 1,054 (18.1%) | 122 (16.1%) | 0.09 |
| 55–74 | 17,874 (44.9%) | 2,588 (44.6 %) | 323 (42.7%) | |
| 75+ | 14,542 (36.5%) | 2,167 (37.3%) | 312 (41.2%) | |
| Region | | | | |
| Lima/Callao | 19,840 (49.8%) | 2,772 (47.7%) | 308 (40.7%) | <0.001 |
| Rest coast | 10,851 (27.3%) | 1,592 (27.4%) | 203 (26.8%) | |
| Amazon | 2,633 (6.6%) | 515 (8.9%) | 66 (8.7%) | |
| Highlands | 6,486 (16.3%) | 930 (16%) | 180 (23.8%) | |
| Facility level | | | | |
| I or II | 17,815 (44.8%) | 2,498 (43%) | 388 (51.3%) | <0.001 |
| III | 21,995 (55.2%) | 3,311 (56.9%) | 369 (48.7%) | |
| ICD-10 | | | | |
| I60 | 3,374 (8.5%) | 529 (9.1%) | 145 (19.2%) | <0.001 |
| I61 | 6,352 (15.9%) | 1,098 (18.9%) | 193 (25.5%) | |
| I63 | 9,635 (24.2%) | 1,643 (28.3%) | 122 (16.1%) | |
| I64 | 20,449 (51.4%) | 2,539 (43.7%) | 297 (39.2%) | |
| LOS (days) [Median (IQR)]–[Range] | 7 (4–13) [2–55] | 7 (4–12) [1–59] | 5 (2–9) [1–53] | <0.001 |

**Notes:**
[a] Only, data time to discharge was included.
[b] Death data were only available for 2016 and 2017.
[c] P-values of intra-hospital mortality obtained through the comparison between categorical variables to the Chi2 test for independence, while this value for numerical variables to the Kruskal-Wallis test.

result, we used a region- and hospital level-stratified Weibull proportional hazard model. We did not include interactions in the region- and hospital level-stratified Weibull proportional hazard model, as the likelihood-ratio test did not display significant differences between models with interactions and without them (P = 0.21 and P = 0.20; P = 0.21 and P = 0.12, respectively for each model).

Subarachnoid hemorrhage (HR = 2.34; 95% CI [1.83–2.99]), intracerebral hemorrhage (HR = 1.89; 95% CI [1.50–2.38]), and stroke not specified as hemorrhage or infarction (HR = 1.48; 95% CI [1.19–1.85]) had higher in-hospital mortality risk compared to ischemic stroke cases. These risk estimates remained similar after adjusting for sex and age: HR = 2.45 (95% CI [1.91–3.14]), HR = 1.95 (95% CI [1.55–2.46]), and HR = 1.45 (95% CI

Based on Kaplan-Meier estimates

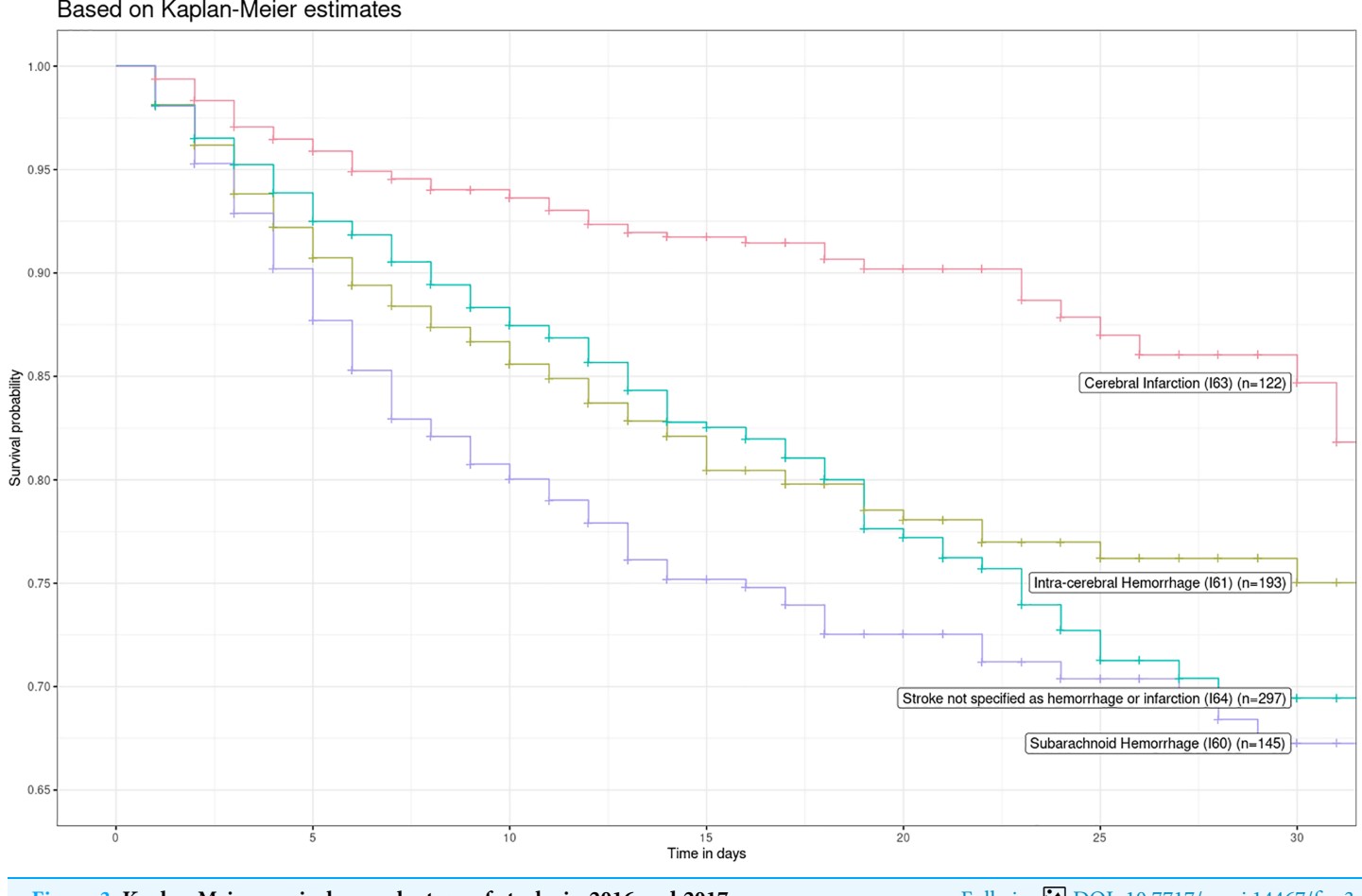

**Figure 3 Kaplan-Meier survival curve by type of stroke in 2016 and 2017.**

[1.16–1.81]) for subarachnoid hemorrhage, intracerebral hemorrhage and stroke not specified as hemorrhage or infarction, respectively.

## DISCUSSION

### Main results

The median LOS of stroke cases was similar in both sexes and has not changed across the years. Hemorrhagic cases had longer LOS than ischemic cases. Between 2016 and 2017, more than 1 in 10 stroke cases died, and after controlling for age and sex, the risk of death was higher among individuals with subarachnoid hemorrhage, intracerebral hemorrhage, and not specified as hemorrhage or infarction, compared to those with ischemic stroke.

### Comparison with previous studies

In our study, we described LOS of stroke patients discharged alive. Different studies have reported median LOS of all stroke patients (alive and dead), showing a higher median LOS among cases with in-hospital stroke for the first time and a similar median LOS among managed stroke cases in the stroke units compared to our findings: a study in Canada, including 28,837 patients with stroke for the first time of community-onset and

intra-hospital onset, reported a median LOS of 8 days and 15 days, respectively (*Saltman et al., 2015*); whereas those cases managed in stroke units had a shorter median LOS of 6 days compared to the median LOS of nine in the stroke cases attended in General Neurology Wards (*Zhu et al., 2009*).

Moreover, LOS seems to depend on the type of stroke, where the ischemic stroke cases have the lowest LOS: a study carried out in the US evaluated hemorrhagic events with the first hospitalization, which reported a longer median LOS (*Reed et al., 2001*) in the subarachnoid hemorrhage cases (8 days), and the same median LOS (5 days) between the intracerebral hemorrhage and ischemic stroke cases. Similarly, a study in Argentina enrolling 1,991 cases of ischemic stroke found a median LOS of 5 days, similar to the one herein reported (*Sposato et al., 2008*), while a Peruvian study conducted in 2,314 stroke patients from one third-level hospital from 2000 to 2009 reported a general median LOS of 7 days consistent with our findings; however, the results of median LOS for each stroke subtype was contrary to those mentioned above: median LOS of ischemic cases were the highest of all with 10 days, followed by hemorrhagic cases (hemorrhagic stroke: 9 days, and subarachnoid hemorrhage: 8 days) and stroke not specified and stroke not specified (8 days) (*Castañeda-Guarderas et al., 2011*).

A systematic review about early CFR (21-day to 1 month) for stroke reported a considerable variation, ranging between 23% and 35% in LMIC from 2000 to 2008, and varied between 20% and 30% in HIC in the same period (*Feigin et al., 2009*). Although data from LMIC was scarce, the incidence of stroke was higher than in HIC, highlighting the need for launching appropriate stroke prevention programs in LMIC paired with robust surveillance or data collection programs. Likewise, a Peruvian study carried out in a third-level facility reported a general CFR of 19.5% in the period 2000–2009, where hemorrhagic stroke had the highest CFR of all with 31.8%, which was slightly higher than our findings (*Castañeda-Guarderas et al., 2011*).

Hemorrhagic strokes (*i.e.*, subarachnoid and intracerebral hemorrhages) had the highest in-hospital mortality risk, as shown in previous studies (*Labberton et al., 2019*; *Deljavan, Farhoudi & Sadeghi-Bazargani, 2018*; *Shoeibi et al., 2015*). A study using data from the Global Burden of Disease 2013, including stroke patients aged 20–64 years, reported two times more deaths in hemorrhagic stroke than ischemic stroke, especially in LMIC (*Krishnamurthi et al., 2015*).

## Results interpretation

This study describes median LOS in stroke patients discharged alive. Despite this, the findings of our study seem to be similar to that of HIC, but there is an increased median LOS in stroke cases managed in specialized centers. Besides, the CFR reported in our study seems to be below international and national estimates. Different factors may play a role in explaining these findings. Inadequate reporting and under-reporting may be potential issues as some stroke patients may not access healthcare or die before accessing the health system, which probably explains the differences between CFR findings (*Nieuwkamp et al., 2009*; *Bryndziar et al., 2020*). The highest frequency of deaths in the first days after hospitalization, mainly due to hemorrhagic events or due to the conditions of

multi-morbidity in the older patients (*Atamari-Anahui et al., 2019*), could lead to a lower LOS, mainly in the elderly patients, as shown in other reports (*Atamari-Anahui et al., 2019*; *Krishnamurthi et al., 2015*). Although some Peruvian clinical practice guidelines for the management of ischemic stroke claim a quick neuroimaging exam (25 min from arrival at the health facility) (*Sequeiros-Chirinos et al., 2020*), it could hardly be carried out in daily clinical practice due to healthcare resource limitations and geographic distances between health facilities to the reference facility, as well as a reduced and centralized number of CT-scan (*Ministerio de Salud, 2020*), among which 26 (46%) are from the capital of Peru, therefore, it could explain the large proportion of stroke cases not specified as hemorrhage or ischemic.

Transferred patients from a first- or second-level to a third-level facility may be severe stroke cases with a higher risk of complications that need ICU care; this could lead to longer LOS and higher CFR (*Reed et al., 2001*; *Nickles et al., 2016*; *Stein et al., 2019*; *Zachrison et al., 2020*), and it may negatively affect the stroke patient outcomes in scarce settings (*Nickles et al., 2016*; *Zachrison et al., 2020*; *Middleton, Grimley & Alexandrov, 2015*; *Détraz, Ernst & Bourcier, 2018*).

Hemorrhagic cases have been noted to have up to 4 times the risk of death than ischemic cases, which is consistent with our findings (*Andersen et al., 2009*). Death outcome seems to be related to stroke severity, type of hemorrhagic lesion, and delay in referral and healthcare access (*Reed et al., 2001*; *Andersen et al., 2009*; *Stienen et al., 2018*; *Qureshi, Mendelow & Hanley, 2009*; *Otite et al., 2017*). Moreover, complicated hemorrhagic events with re-bleeding require adequate management as soon as possible (*Stienen et al., 2018*). As such, individuals with hemorrhagic stroke are at high risk of death, given the shortcomings mentioned above in the Peruvian health system.

## Public health relevance

LOS is an important indicator that reflects the use of hospital resources. In Peru, LOS in stroke patients has been similar over the years and relatively similar to LOS of LMIC and HIC. However, these results must be interpreted with caution due to the heterogeneous characteristics between the populations. Thus, a mortality report from the Ministry of Health indicated a growing stroke mortality trend in the period 2003–2015 (*Ministerio de Salud, 2018*), while a Peruvian study with data from the Ministry of Health reported global stroke mortality decreased in the period 2005–2015 (*Atamari-Anahui et al., 2019*), which can be explained by the different methodologies used in both studies. In this study, the mortality trend could not be determined because we did not have mortality data for that period. In local clinical experience, Peru faces challenges related to timely access to stroke care; these challenges are much more profound in settings with limited health resources, where the stroke units are limited, and instead, there are second- and third-level facilities that attend patients with acute stroke, of which third-level facilities can provide access to CT-scan (*Sequeiros-Chirinos et al., 2020*; *Ministerio de Salud, 2020*). These problems require urgent attention through strategies to improve access to healthcare (*i.e.*, telediagnosis and telemonitoring) and decentralization strategies (*i.e.*, the creation of stroke units with neuroimaging) to reduce hospital resource gaps (*Jaillard, Hommel &*

*Mazetti, 1995*; *O'Brien et al., 2017*; *Tamm et al., 2014*). This way, it could make the diagnosis and management more effective and less expensive for the families of stroke patients. Also, there is a need to monitor appropriate stroke care indicators and assess the impact of potential interventions (*Ohinmaa et al., 2016*).

Public health policies should be strengthened and aimed to reduce the prevalence and mortality related to stroke. These objectives can be reached by the decentralization of both human and logistic resources, along with a reorganization of the healthcare system, so that patients can receive adequate primary or secondary prevention without exhausting the limited resources (*Audebert & Sobesky, 2014*; *Fulop et al., 2019*), and by increasing health literacy in the general population; thus, they can acknowledge the relevance of identifying early signs of stroke to seek healthcare are promptly (*Feigin et al., 2009*; *Stienen et al., 2018*; *Emberson et al., 2014*).

### Strengths and limitations

Benefiting from a large national database, we studied trends of in-patient LOS and in-hospital mortality among stroke patients in Peru. However, this study presents some limitations that should be highlighted. First, we only utilized records from the MoH, which provides health care to about 55% of the population, especially those with economic limitations. Therefore, we consider that the distribution of the population included in our study is representative of those insured to MOH. In addition, the studied sample is sufficiently large, which optimizes the generalizability of our results to the Peruvian population affiliated to MOH. However, the socioeconomic status of those insured to MOH would not support generalizing our results to the private sector and other countries. Second, we could not control whether patients were referred from the first- and second-level facilities to third-level facilities; besides, we could not identify which patients were readmitted for stroke as these data were not available nor data on stroke severity were available, so our data could not be representative for populations of mild-to-moderate stroke admissions who survive and spend a short LOS. Third, mortality data were provided only from 2016 to 2017, and unmeasured confounders (lack of data regarding stroke severity measurements (*i.e.*, the NIHSS scale), disability at discharge (*i.e.*, the modified Ranking Scale), prestroke medications, acute stroke treatments, ICU or operating room utilization and complications associated with hospitalizations and interventions (*i.e.*, hospital-acquired infections), and delay in arrival at the reference health center) could have biased our estimates. Fourth, there was no availability of data related to stroke hospital costs, so it was not possible to get complementary estimates of the use of health resources in relation to LOS. Finally, both coding bias and lack of information coverage in regions outside large cities could underestimate reported cases.

### CONCLUSIONS

LOS in patients with stroke has not changed in Peru from 2002 to 2017, consistent with previous studies in Peru and other LMIC and HIC. Hemorrhagic stroke had longer LOS than ischemic stroke. Hemorrhagic strokes (*i.e.*, subarachnoid and intracerebral

hemorrhage) had a higher risk of in-hospital mortality. Our findings suggest the need for strategies to improve access to stroke care.

### Funding
This work was supported by a Wellcome Trust International Training Fellowship grant number (214185/Z/18/Z). The funders had no role in study design, data collection and analysis, decision to publish, or preparation of the manuscript.

### Grant Disclosures
The following grant information was disclosed by the authors:
Wellcome Trust International Training Fellowship: 214185/Z/18/Z.

### Competing Interests
The authors declare that they have no competing interests.

### Author Contributions
- L. Max Labán-Seminario performed the experiments, analyzed the data, prepared figures and/or tables, authored or reviewed drafts of the article, and approved the final draft.
- Rodrigo M. Carrillo-Larco conceived and designed the experiments, performed the experiments, prepared figures and/or tables, authored or reviewed drafts of the article, and approved the final draft.
- Antonio Bernabé-Ortiz conceived and designed the experiments, performed the experiments, prepared figures and/or tables, authored or reviewed drafts of the article, and approved the final draft.

### Data Availability
Analysis code is available through GitHub:
https://github.com/llabans/stroke.

The dataset is available at Figshare: Labán-Seminario, L. Max; Bernabe-Ortiz, Antonio; Carrillo Larco, Rodrigo (2020): PERU STROKE study. figshare. Dataset. https://doi.org/10.6084/m9.figshare.12552011.v3.

### Supplemental Information
Supplemental information for this article can be found online at http://dx.doi.org/10.7717/peerj.14467#supplemental-information.

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
