# Peer review of "Stroke-related length of hospitalization trends and in-hospital mortality in Peru"

_PeerJ, doi:10.7717/peerj.14467_

## Round 0.1 · original submission · Major Revisions

Both reviewers expressed serious concerns regarding the clarity of the study design, reporting of the results and appropriate citation choices. It will be important to carefully address the issues raised.

Reviewer 1 ·

Basic reporting

Basic reporting: this manuscript is written in clear and unambiguous professional English. However, it requires an improvement in certain areas of sentence construction.
The background points towards the need of understanding the public health structure and policy regarding the care of strokes in Peru. The relationship between length of stay and the growth of the health system for the management of stroke is not real. I don’t think that outcome can be the surrogate outcome to show the response of the health system.
The graphics fig 2 need to have legend in Y axis; Kaplan Meier should not have the shadows just lines it gets too colorful and overwhelming
Background sentence line 83-87 needs to be re-written, long and does not give a clear idea of what they are trying to explain.

Experimental design

Experimental design: Question is not clearly stated. There is no clear hypothesis. There are two aims, one is the study with LOS and the other one is the evaluation of the type of stroke to mortality as outcome.
Methods, study design there is no statement of the design, in abstract they mention a cohort study.
It is not clear if this is public data or how it was shared to investigators
The ICD-10 codes were added just with the first numbers, no decimal numbers specified. Did that capture all the patients with stroke?
Errors in analysis, for numerical they need to specify it is continuous. Chi-square no “d” at the end.
LOS can be analyzed with a linear logarithmic model and mortality with a logistic regression.

Validity of the findings

Results are presented appropriately.
The results of this study are not appropriately discussed. The difference between regions seems to be more relevant to the public health discussion. The study does not include clinical variables to mention if protocols are appropriate or not or if the availability of CT scan changes or not in centers where this data was analyzed. There is no discussion regarding costs. There is no discussion regarding management, regarding ICU utilization, OR utilization, teams involved. All this could also be mentioned in limitations as drawing conclusions different that what the numbers show should not be done. They mention the LOS is similar to the one reported by other countries so why is the findings of LOS makes them say that these results reveal challenge the Peruvian health system regarding stroke care. There must be something that is being done appropriately as well. Better answers they could obtain by showing how many I II and III-ary hospitals they have in each region selected and what are the distances to the hospitals capable to treating this diagnosis. Having a catchment area and economical production and budget for health for those regions could be more helpful. I recommend looking at the methods of cited study 40 “Healthcare Resource Availability, Quality of Care, and Acute Ischemic Stroke Outcomes” Emily C. O'Brien , PhD emily.obrien@duke.edu , et al.
Conclusion needs to be improved with recommendations from discussion section

Additional comments

Abstract needs to be more concise and include recommendations in this review.

·

Basic reporting

- Authors aimed to report their results by stroke type. This is relevant; however, in the Introduction section, there is no information about how types of stroke might differentially impact the burden of disease, costs, and patients' hospitalizations and deads.
- I suggest explaining the sentence in lines better 87-90 "Studying Length of stay (LOS) trends might give evidence(...)thereby, we can identify where interventions are most needed". Because it is not clear how the LOS could give information about the response of a health system if this last indicator is quite more complex. In addition, how it might be possible to identify the needs of interventions only with LOS indicators.
- References 4, 12, 40, 41 are in a different Vancouver format citation (As web page, not as a research paper from an academic journal). Please correct them.
- References 11, 12, 17, 18, 30 are in Spanish. I suggest translating into English using brackets ([]) to get a better references style for all readers.
- I suggest that the authors consider discussing or add in the introduction section these other references which had similar outcomes to yours. Stroke in local public hospitals (https://scielosp.org/article/ssm/content/raw/?resource_ssm_path=/media/assets/rpmesp/v28n4/a08v28n4.pdf) (http://www.cimel.felsocem.net/index.php/CIMEL/article/view/11/17).
- There is a previous report of the general national mortality for stroke in Peru during 2005-2015 using a similar dataset as the authors (From the Peruvian ministry of health) (https://www.sciencedirect.com/science/article/abs/pii/S1853002819300461). However, the authors didn't mention it as relevant previous research. Therefore, I suggest better bibliographic research to better understand these findings in the local context.

Experimental design

- I suggest an explanation for the reason only to include adults aged 35 years or older in the analysis.
- There is a reason why the authors categorized the age in (35-54, 55-74, and ≥75 years)? Please, provide it.
- The Mann-Whitney-Wilcoxon Rank Sum tests are not used to compare between two numerical variables. Instead, they test the equality of means or ranges between two (independent or dependant) samples. Please correct or provide further explanation. Also, why the authors used a hypothesis test for paired or dependant groups?. Maybe for the limitation of possible duplicate reports because of patient readmission. However, this has to be specified in the methods section.
- There is a considerable number of excluded records outside the 1st and 99th percentile (n=3573). I suggest mentioning the range of LOS in these records to understand their characteristics better.

Validity of the findings

I have some concerns about the validity of the database because:
- 58% of all hospitalization records are only from stroke patients? Is this possible? or it might be some bias of the database.
- The authors excluded registries with a "discharge with death registry" from the 2002-2017 database, so I assume that these 892 excluded registries (according to Fig 1) are only from 2016 and 2017. Please further explain this situation. However, suppose these 892 registries are from the 2002-2017 database. In that case, I will find it quite low for in-hospital mortality during 15 years (Very different to more than 4000 stroke death reported in a previous paper (Atamari-Anahui, 2019). And also, why did the authors analyze in-hospital mortality only for 2016-2017 if the remaining records had "discharge with death registry"?.
- If the authors excluded registries with in-hospital stroke death only from 2016 and 2017, they analyzed the LOS of patients who lived and died during 2002-2015 and only the patients who lived during 2016-2017. This will create a severe bias that may alter all their results.
- In table 1, there were 757 deaths during 2016-2017, but the authors excluded 892 registries because of the "discharge with death registry" (according to Fig 1). Please explain this difference.

Additional comments

- During the discussion, I suggest to the authors to specify if similar studies are measuring LOS in all hospitalized patients (live or death) or only in the ones who survived to stroke. The same in the case of deaths, please specify if they are measuring in-hospital dead or general dead, because this might be the main reason to get a lower CFR than other countries.

---

## Round 0.2 · Minor Revisions

Both reviewers feel there have been substantial improvements to the paper. Reviewers have some disagreement regarding the reference Labán (2021) [21]. I agree with Reviewer #2 that the Labán (2021) [21] citation should be included but the methods from that citation synthesized and included in this manuscript. Please make the other minor edits as requested.

Reviewer 1 ·

Basic reporting

Language can be improved thoughout the paper.
Improvements made helping to understand results.

Experimental design

Improved.
I would refrain from citing the study by Luis Laban in methods and results. That reference is not a peer reviewed journal publication. I believe you have a good explanation of what are the methods without writting or citing methods by someone else in paper.

Validity of the findings

Valid findings. Improved reporting. Conclusion can be cleaned. I don't know if your data supports concluding that "Our findings suggest the need for promoting adequate stroke prevention programs and decentralization of health resources to improve access to stroke care, the identification and management of stroke cases." Your LOS was similar to other references. You would need an epidemiological study to look at that.

Additional comments

Improved manuscript. I suggest having a correction of language to improve redaction and scientific writting.

·

Basic reporting

- I suggest adding your explanations in the methods section about the exclusion of adults aged 35 or older (and your sensitivity analysis in results sections, if any)
- If the journal had no issues with the length of the manuscript and the number of words, I suggest that the authors synthesize the methods in "Luis Labán (2021) [21]" into the methods section of the manuscript.

Experimental design

- As mentioned in the methods section, during the analysis of LOS 2002-2017, the authors excluded the patients with "discharge with death registry." How is this last criterion different from in-hospital mortality?

Validity of the findings

- I still have concerns trying to understand what kind of population was included in the original database (n=98,605). As mentioned in "Luis Labán (2021) [21]", did the authors ask the MoH for the registries of all hospitalized patients with any medical diagnosis in any MoH health facility or for the registries with only ICD-10 codes for stroke? (this should be acknowledged in the methods section). If the authors used all registries with any diagnosis, 58% of all these registries are too high, considering that the incidence of stroke during 2017 (according to Bernabe-Ortiz (2021)) was 33.2 per 100,000 person-year. Otherwise, if the authors originally used only the records with ICD-10 codes for stroke, why did they exclude 42% of the records as "not-stroke cases" as mentioned in their results section "Of these, 58% (n=57,153) were stroke cases"?.

- If "stroke deaths were only available in 2016 and 2017", the authors did combine in the LOS analysis two kinds of registries: a) patients registered during 2002-2015 with no data about dead (authors are assuming that all were discharged alive?) and b) patients who lived during 2016-2017.? This combination could create bias.

Additional comments

- Due to the different criteria for analyses, for a better understatement, I suggest providing two different flowcharts for the LOS analysis and the in-hospital mortality analysis.

---

## Round 0.3 · accepted · Accept

The authors have adequately addressed the concerns raised by the reviewers.The reviewers expressed only minor concerns which I assessed in this revision. The manuscript is ready for publication.